# Seroprevalence of anti-SARS-CoV-2 specific antibodies in vaccinated and vaccine naïve adult Nigerians

Abdulfattah Adekunle Onifade[1], Adeola Fowotade[2,3], Sheu Kadiri Rahamon👤[1]*, Victory Fabian Edem[1], Surajudeen Adebayo Yaqub👤[1], Olatunji Kadri Akande[2], Olatunbosun Ganiyu Arinola[1]

1 Department of Immunology, College of Medicine, University of Ibadan, Ibadan, Nigeria, 2 Biorepository Clinical Virology Laboratory, College of Medicine, University of Ibadan, Ibadan, Nigeria, 3 Department of Medical Microbiology, College of Medicine, University of Ibadan, Ibadan, Nigeria

* adekunlesheu@rocketmail.com, sk.rahamon@mail1.ui.edu.ng

## Abstract

### Background

Reports on the evaluation of immune responses to different COVID-19 vaccines are limited. Similarly, effects of age and gender have not been well explored as variables that could impact on the vaccine-induced antibody response. Therefore, seroprevalence of anti-SARS-CoV-2 specific antibodies in vaccinated and vaccine naïve adult Nigerians was determined in this study.

### Methodology

A total of 141 adults were enrolled into this study. Presence or absence of SARS-CoV-2 infection was confirmed by real-time reverse-transcriptase polymerase-chain reaction (RT-PCR) assay on nasopharyngeal and oropharyngeal swab specimens. Anti-SARS-CoV-2 Specific IgG and IgM antibodies were qualitatively detected using a Rapid Diagnostic Test kit.

### Results

Pre-vaccination, 77% of the study participants had never had PCR-confirmed COVID-19 test yet 66.7% of them were seropositive for SARS-CoV-2 antibodies. Of 111 COVID-19 vaccinated participants, 69.2% and 73.8% of them had SARS-CoV-2 specific IgG post-first and second doses of COVID-19 vaccine respectively. However, 23.1% and 21.4% of the participants who have had first and second doses respectively had no detectable anti-SARS-CoV-2 antibodies. The proportion of participants with SARS-CoV-2 specific IgG was insignificantly higher in those between the ages of 18–40 years and 41–59 years compared with individuals aged ≥60 years. No significant association was observed between gender and seropositivity for SARS-CoV-2 antibodies.

### Conclusion

There is high SARS-CoV-2 antibody seroprevalence among Nigerian adults who never had PCR-confirmed COVID-19. Also, there is the need for anti-SARS-CoV-2 antibodies

**Data Availability Statement:** All relevant data are within the manuscript.

**Funding:** The author(s) received no specific funding for this work.

**Competing interests:** The authors have declared that no competing interests exist.

screening post vaccination as this could be essential in achieving herd immunity. Age and gender do not seem to have significant association with seropositivity.

## Introduction

Coronavirus disease 2019 (COVID-19) caused the highly infectious Severe Acute Respiratory Syndrome Coronavirus 2 (SARS-CoV-2) continues to be an unprecedented global health crisis [1]. Its associated morbidity and mortality resulted in the rapid development of diverse SARS-CoV-2 vaccines which are currently in use while others are still being developed or in different phases of clinical trials.

As of the 18th of March 2021, about thirteen COVID-19 vaccines had been approved for use in different countries while several others were at different phases of randomized clinical trials [2]. Interestingly, many more are still emerging with a view to improving on efficacy especially against the emerging variants of SARS-CoV-2 [3].

At present, there is no generally accepted specific treatment for COVID-19 hence, vaccination remains the most important fulcrum for prevention of the disease [4,5]. The mechanism of action of COVID-19 vaccine, just like many other vaccines, are either based on active immunity (e.g. live attenuated, viral vector and DNA/RNA vaccines) or passive immunity (e.g. monoclonal/polyclonal antibodies) [6].

Although substantial progress has been made in vaccine development, concerns about safety and effectiveness are still challenges which require further research. The report of Irwin and Nkengasong showed that 70% of all humans must be vaccinated to eradicate COVID-19 [7]. In Nigeria, the Nigeria Centre for Disease Control (NCDC) aimed to vaccinate 40% of the Nigerian population and hopes to achieve the 70% vaccination threshold for eliminating COVID-19 before the end of the year 2022 [8]. As of 7 June 2022, a total of 11,854,673,610 vaccine doses have been administered globally [9]. In Nigeria, as of 29 May 2022, about 30,680,510 (14.9% of the population) Nigerians have taken at least 1 dose while 20,096,868 (9.7% of the population) have taken 2 doses and are thus, fully vaccinated [10]. Thus, the uptake of COVID-19 vaccine remains low in Nigeria.

Avalanche of reports on the host immune response to COVID-19 and advances in molecular techniques have facilitated rapid development of COVID-19 vaccines. However, there is the dearth of information on possible differences between SARS-CoV-2 infection-induced immunity and vaccine-induced immunity. It is therefore of clinical importance to determine the antibody responses in vaccinated and unvaccinated individuals in order to determine the possibility of achieving herd immunity. The primary antigens for antibody production during natural SARS-CoV-2 infection are the spike (S) and nucleocapsid (N) proteins [11]. Following infection with SARS-CoV-2; naïve B cells are activated via antigen recognition and CD4$^+$ T cells activation. This activation results in cascade of events which leads to production of antibodies and memory B cells.

Available reports showed that majority of SARS-CoV-2 patients seroconvert shortly after onset of symptoms with viral-specific IgG, IgA and IgM developing simultaneously [12–16]. However, this seroconversion could occur in phases; IgM seroconversion earlier than IgG, IgG seroconversion earlier than IgM and synchronous seroconversion of IgM and IgG [12,17,18]. In addition, the median time to seroconversion varies [19]. Iyer *et al.* [17] and Long *et al.* [12] reported that that the median time to seroconversion is between 11 and 13 days post-symptom onset (PSO). Also, Röltgen *et al.* [18] reported that the seroconversion rates in hospitalized patients for anti-S receptor binding domain (RBD) IgM, IgG and IgA reached their maximum

between 4 to 6 weeks PSO. It is therefore not surprising that changes in the dynamics of SARS-CoV-2-specific antibodies as well as memory B and T cells have been well explored in the design of various vaccines against the infection and guided the advocacy for booster vaccines. Studies have shown that COVID-19 patients who are symptomatic develop SARS-COV-2-specific antibodies [20] and waning begins after few months [21].

Although the concentrations of these SARS-CoV-2 specific antibodies following natural infection or vaccination are crucial in determining the roles of these antibodies as either protective or disease facilitating antibodies [19,22–25], data on the antibody titre threshold that will confer optimal protection is limited.

Furthermore, there is the dearth of information on seroprevalence of anti-SARS-CoV-2 specific antibodies in vaccinated and vaccine naïve individuals. Availability of this information is of clinical importance as it could be essential in assessing the possibility of achieving herd immunity and could further establish the use of sero-epidemiological reports in developing immunization policies. These thus serve as the basis for this study.

## Materials and methods

### Study site

The study participants were consecutively enrolled from adults seeking COVID-19 screening at the Biorepository Laboratory, College of Medicine, University of Ibadan and the general populace in Ibadan North Local Government Area of Oyo State, Nigeria.

### Study population

A total of 141 adults were enrolled into this study. Participants with past SARS-CoV-2 infection history were those confirmed positive by real-time reverse-transcriptase polymerase-chain reaction (RT-PCR) assay on nasopharyngeal and oropharyngeal swab specimens following the World Health Organization (WHO) guideline [26]. All the vaccinated participants had their vaccination at the Nigeria Centre for Disease Control (NCDC) approved centres and were enrolled into the study after at least, one month post vaccination.

### Exclusion criteria

Participants with infections such as HIV or pulmonary tuberculosis were excluded from this study.

**Ethical consideration.** The study was conducted after obtaining approval from the University of Ibadan/University College Hospital (UI/UCH) Joint Ethics Review Committee (UI/EC/20/0233). Written informed consent was also obtained from the study participants.

### Determination of SARS-CoV-2 specific IgM and IgG antibodies

SARS-CoV-2 specific antibodies were qualitatively detected using a commercially available Rapid Diagnostic Test kit (Wuhan UNscience Biotechnology Company Limited, China). The kit works on the principle of lateral flow immunoassay and detects IgG and IgM antibodies to SARS-CoV-2 in human whole blood, serum, or plasma. It has a clinical sensitivity of 98.511% (95% CI: 96.788% - 99.452%) and specificity of 88.208% (95% CI: 83.086% - 92.221%). The detection was carried out following the manufacturers' instruction.

### Statistical analysis

Statistical analysis was carried out using SPSS statistical software version 23.0 for windows. Chi-square test or Fischer's exact test, as appropriate, was used to determine the association

**Table 1. Gender and age of the study participants.**

|  | Number | Percentage (%) |
|---|---|---|
| Gender |  |  |
| Male | 57 | 40.4 |
| Female | 84 | 59.6 |
| Age (years) |  |  |
| ≤ 40 | 71 | 50.4 |
| 41–59 | 49 | 34.8 |
| ≥60 | 21 | 14.9 |

between categorical variables. *P*-values less than 0.05 were considered as statistically significant.

## Results

The gender and age groups of the study participants are shown in Table 1. The study participants were largely female and more than 50% of them were not older than 40 years.

As shown in Table 2, about 77% of the study participants were unsure of their COVID-19 history as they had never had COVID-19 PCR test done. The proportion of participants who had either the first or second dose of COVID-19 vaccine was higher than those who were not yet vaccinated. Over 90% of the participants had AstraZeneca vaccine administered (Table 2). Coincidentally, 7 (77.8%) of the 9 participants that took Moderna vaccine had detectable SARS-CoV-2 antibodies while 2 (22.2%) did not. Similarly, 77 (77.8%) of the 99 participants that took Astrazeneca vaccine had detectable SARS-CoV-2 antibodies while 22 (22.2%) did not.

Responses to COVID-19 vaccine are contained in Table 3. Twenty (60.6%) participants who were not yet vaccinated were seropositive for SARS-CoV-2-specific IgG while 6.1% were seropositive for both SARS-CoV-2-specific IgG and IgM. As expected, post first and second doses of COVID-19 vaccine, 69.2% and 73.8% of the study participants had SARS-CoV-2 specific IgG. However, 7.7% of those who had taken first dose of the vaccine had a combination of IgM and IgG. Unexpectedly, 23.1% and 21.4% of the participants who have had first and second doses respectively, had no protective antibody to SARS-CoV-2. The only participant who has had the booster dose was seronegative (Table 3).

**Table 2. History of COVID-19 and vaccine uptake in the study participants.**

|  | Number | Percentage (%) |
|---|---|---|
| COVID-19 History |  |  |
| Previous PCR-confirmed infection | 17 | 12.1 |
| No history of PCR-confirmed infection | 15 | 10.6 |
| Never had COVID-19 test | 109 | 77.3 |
| Vaccine History |  |  |
| Not yet vaccinated | 33 | 23.4 |
| Post first dose | 65 | 46.1 |
| Post second dose | 42 | 29.8 |
| Post booster dose | 1 | 0.7 |
| Vaccine Type |  |  |
| Astrazeneca | 99 | 91.7 |
| Moderna | 9 | 8.3 |

**Table 3. Association between seropositivity and vaccine history.**

| | Not yet vaccinated | PFD | PSD | PBD | N | $X^2$ | P-value |
|---|---|---|---|---|---|---|---|
| Seronegative | 11 (33.3%) | 15 (23.1%) | 9 (21.4%) | 1 (100.0%) | 36 | 12.459 | 0.189 |
| IgM | 0 (0.0%) | 0 (0.0%) | 2 (4.8%) | 0 (0.0%) | 2 | | |
| IgG | 20 (60.6%) | 45 (69.2%) | 31 (73.8%) | 0 (0.0%) | 96 | | |
| IgM + IgG | 2 (6.1%) | 5 (7.7%) | 0 (0.0%) | 0 (0.0%) | 7 | | |

PFD = Post first dose, PSD = Post second dose, PBD = Post booster dose, n = Number, IgG = Immunoglobulin G, IgM = Immunoglobulin M.

**Table 4. Association between seropositivity and age groups.**

| | 18–40 years | 41–59 years | ≥60 years | N | $X^2$ | P-value |
|---|---|---|---|---|---|---|
| Seronegative | 20 (28.2%) | 9 (18.4%) | 7 (33.0%) | 36 | 9.570 | 0.144 |
| IgM | 0 (0.0%) | 2 (4.1%) | 0 (0.0%) | 2 | | |
| IgG | 50 (70.4%) | 34 (69.4%) | 12 (57.1%) | 96 | | |
| IgM + IgG | 1 (1.4%) | 4 (8.2%) | 2 (9.5%) | 7 | | |

n = Number, IgG = Immunoglobulin G, IgM = Immunoglobulin M.

**Table 5. Association between seropositivity and gender.**

| | Male | Female | N | $X^2$ | P-value |
|---|---|---|---|---|---|
| Seronegative | 15 (26.3%) | 21 (25.0%) | 36 | 3.407 | 0.333 |
| IgM | 2 (3.5%) | 0 (0.0%) | 2 | | |
| IgG | 38 (66.7%) | 58 (69.0%) | 96 | | |
| IgM + IgG | 2 (3.55) | 5 (6.0%) | 7 | | |

n = Number, IgG = Immunoglobulin G, IgM = Immunoglobulin M.

In Table 4, the association between age and seropositivity is shown. The proportion of participants with SARS-CoV-2 specific IgG was insignificantly higher in those between the ages of 18–40 years and 41–59 years compared with ≥60 years.

Considering the association between gender and seropositivity, the proportion of male participants with SARS-CoV-2 specific IgM or IgG was not significantly different from the proportion of female participants with SARS-CoV-2 specific IgM or IgG (Table 5).

## Discussion

The epidemiology of SARS-CoV-2 infection and its associated morbidities and mortality vary worldwide. In Nigeria, the infection and mortality rate has been relatively lower compared with many developed countries and even, some African countries. As at 27th May, 2022, the total number of confirmed cases was 256,004 while a total of 3,143 deaths were reported [27]. Reasons for these observations are poorly understood but are of epidemiological importance.

The high number of individuals who never had COVID-19 test observed in this study corroborates the report of Al-Mustapha *et al.* [28]. These observations showed that the true prevalence of SARS-CoV-2 infection is probably not known in Nigeria and by extension, in many African countries. The NCDC report of May 27, 2022, showed that 5,160,280 Nigerians have, so far, been tested for SARS-CoV-2 infection [27]. Our observed poor COVID-19 testing could be due to the myriads of challenges inundating the health system in Nigeria culminating

in sub-optimal services [29]. More importantly, a number of challenges are associated with molecular testing for SARS-CoV-2.

In poor resource countries, molecular testing is largely available in reference laboratories; this may result in poor access to the testing. Also, the use of molecular testing for prompt clinical decision is precluded by prolonged turnaround time. The cost of molecular testing is another important factor which could pose impediment to mass testing in low and middle income countries [30–32]. Although, in Nigeria, community SARS-CoV-2 testing is free and molecular testing capacity has expanded significantly, lack of interest in voluntary testing in asymptomatic patients and the need for isolation, in confirmed patients, which restricts daily activities resulting in dire economic consequences still constitute significant challenge to mass testing. These non-health system related factors could be responsible for our observation.

Serological surveys have been reported to be the best tool to determine the spread of an infectious disease [33]. The reports of Byambasuren et al. [34] and Sughayer et al. [35] showed that the true magnitude and spread of SARS-CoV-2 infection can be accurately reflected through seroprevalence estimates. The high proportion (66.7%) of individuals who never had COVID-19 vaccine but were seropositive for SARS-CoV-2 observed in this study supports the report of George et al. [36] which showed an overall seroprevalence of 62.7% in 831 vaccine naïve adults in a rural district of South India. Similarly, Murhekar et al. [37] reported a SARS-CoV-2 seroprevalence of 67.6% in 28,975 Indian population aged ≥6 years. They also showed that the seroprevalence was not different in rural or urban areas. Our observation, together with the previous reports, showed that the true magnitude of SARS-CoV-2 infection could be better estimated through sero-epidemiological studies. In addition, observation from this study suggests that there is the need for SARS-CoV-2 specific antibody screening before vaccination in order to optimize available vaccine doses especially in poor resource countries that largely depend on vaccine donations. The use of serology tests and sero-epidemiological reports in supporting immunization policies against vaccine-preventable diseases is well established [38].

Antibody levels are a relatively crude tool to assess vaccine effectiveness. However, the durability of immune protection following vaccination as compared to natural infection remains an important question [39]. Anti-SARS-CoV-2 antibodies have been shown to be convenient surrogate in both vaccinated and naturally immunized individuals [40–42]. Murhekar and colleagues [37] reported seroprevalence rates of 81% and 89.8% among adults who had received one dose and 2 doses of COVID-19 vaccines respectively. Similarly, Hamaya et al. [43] reported seroprevalence rate of 98% among healthy adults after the second dose of the BNT162b2 mRNA vaccine. Our study is in tandem with these previous studies as 69.2% and 73.8% seroprevalence rates were observed post first and second doses of COVID-19 vaccine respectively. It could therefore be stated that more than 50% of Nigerians who had COVID-19 vaccine produced antibodies against SARS-CoV-2.

Comparing the efficacy of vaccines head-to-head is practically impossible due to a number of factors. Some of these factors include approaches taken in designing the studies, different endpoint of clinical trials, and differing case rates and circulating variants during the period of the clinical trials among others [44,45]. Reports have shown that the AstraZeneca vaccine has about 76% efficacy against symptomatic SARS-CoV-2. This efficacy is specific to events from 15 days post second-dose, with an interval dose of 29 days [45]. Similarly, the Moderna vaccine has been reported to have about 94.1% efficacy against the ancestral strain of SARS-CoV-2, starting from day 14 post first-dose [46]. These reports probably explain our observation that 23.1% and 21.4% of the participants who have had first and second doses of COVID-19 vaccine respectively had no detectable anti-SARS-CoV-2 antibodies. More worrisome was our observation that the only participant who has had the booster dose also had no detectable anti-

SARS-CoV-2 antibodies. The report of Stringhini *et al.* [47] showed that 3 adults who have had at least one dose of COVID-19 vaccine had no detectable anti-SARS-CoV-2 antibodies after more than 14 days of vaccination. These observations are of clinical importance as they suggest that there is the need for anti-SARS-CoV-2 antibodies screening post vaccination to identify cases of vaccine escape and to ensure that protection is achieved. Such screening is necessary if herd immunity is to be achieved. In addition, our observation could further establish that SARS-CoV-2 vaccine-induced antibodies wane over time.

In this study, majority of the participants were vaccinated with AstraZeneca (91.7%) compared with Moderna (8.3%). This observation could be due to availability. A large mass of COVID-19 vaccines administered in Nigeria were donated by several donor agencies and developed countries. It could also be due to false believes of relative severities of adverse reactions. Interestingly, results from our study showed that participants who had Astrazeneca vaccine had similar antibody response with those that took Moderna vaccine. Seven (77.8%) of the 9 participants that took Moderna vaccine had detectable SARS-CoV-2 antibodies while 2 (22.2%) did not. Similarly, 77 (77.8%) of the 99 participants that took Astrazeneca vaccine had detectable SARS-CoV-2 antibodies while 22 (22.2%) did not.

Seroprevalence estimates have been shown to differ significantly across age groups. It was shown to be lowest among children and highest among older adults [47]. Similarly, Murhekar *et al.* [37] reported that seroprevalence post-vaccination increased with age. In our study, a dissimilar pattern was observed as the proportion of participants with SARS-CoV-2 specific IgG was insignificantly higher in those between the ages of 18 and 40 years, and 41 and 59 years compared with those aged ≥60 years. Our observation might indicate that the immune response to the antigen contained in the vaccine is relatively lower in the elderly compared with the young adults; this possibility is in line with the immunosenescence hypothesis.Our observation is consistent with several reports which showed that relatively older COVID-19 vaccine recipients had a diminished IgG response as compared to younger recipients [48–54].

Gender was explored as a possible modifier of the vaccine response. We observed no significant association between gender and seropositivity. Keshavarz et al. [54] reported that lower IgG levels were not significantly associated with sex in recipients of BNT162b2 at day 7–20 post-boost.

At present, there is the dearth of information on the antibody titre threshold, induced via vaccination or natural infection, which could be considered adequate to protect an individual from SARS-CoV-2 infection. Longitudinal scale studies to determine the presence of neutralizing antibodies in vaccinated individuals are recommended.

It could be concluded from this study that there is high SARS-CoV-2 antibody seroprevalence among Nigerian adults who never had any form of COVID-19 testing. This further confirms the erstwhile suspicion of large-scale undetected community transmission. Also, it is of clinical importance to screen for anti-SARS-CoV-2 antibodies post vaccination as it could be essential in achieving herd immunity. Age and gender do not seem to have significant association with seropositivity. Information from this study could help policymakers make informed decisions regarding vaccination strategies in Nigeria.

## Author Contributions

**Conceptualization:** Abdulfattah Adekunle Onifade, Adeola Fowotade, Sheu Kadiri Rahamon, Olatunbosun Ganiyu Arinola.

**Data curation:** Adeola Fowotade, Sheu Kadiri Rahamon, Victory Fabian Edem, Surajudeen Adebayo Yaqub, Olatunji Kadri Akande, Olatunbosun Ganiyu Arinola.

**Formal analysis:** Sheu Kadiri Rahamon.

**Investigation:** Abdulfattah Adekunle Onifade, Adeola Fowotade, Sheu Kadiri Rahamon, Victory Fabian Edem, Surajudeen Adebayo Yaqub, Olatunji Kadri Akande, Olatunbosun Ganiyu Arinola.

**Methodology:** Abdulfattah Adekunle Onifade, Adeola Fowotade, Sheu Kadiri Rahamon, Victory Fabian Edem, Surajudeen Adebayo Yaqub, Olatunji Kadri Akande, Olatunbosun Ganiyu Arinola.

**Project administration:** Abdulfattah Adekunle Onifade, Sheu Kadiri Rahamon, Olatunbosun Ganiyu Arinola.

**Software:** Sheu Kadiri Rahamon.

**Supervision:** Olatunbosun Ganiyu Arinola.

**Validation:** Abdulfattah Adekunle Onifade, Adeola Fowotade, Olatunbosun Ganiyu Arinola.

**Writing – original draft:** Abdulfattah Adekunle Onifade, Adeola Fowotade, Sheu Kadiri Rahamon, Victory Fabian Edem, Surajudeen Adebayo Yaqub, Olatunji Kadri Akande, Olatunbosun Ganiyu Arinola.

**Writing – review & editing:** Abdulfattah Adekunle Onifade, Adeola Fowotade, Sheu Kadiri Rahamon, Victory Fabian Edem, Surajudeen Adebayo Yaqub, Olatunji Kadri Akande, Olatunbosun Ganiyu Arinola.

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
