## [Decision Letter · Decision Letter 0]

9 Aug 2022

PONE-D-22-18914Seroprevalence of anti-SARS-CoV-2 specific antibodies in vaccinated and vaccine naïve adult NigeriansPLOS ONE

Dear Dr. Rahamon,

Thank you for submitting your manuscript to PLOS ONE. After careful consideration, we feel that it has merit but does not fully meet PLOS ONE’s publication criteria as it currently stands. Therefore, we invite you to submit a revised version of the manuscript that addresses the points raised during the review process.

ACADEMIC EDITOR: The authors are requested to address the issues raised by one of the reviewer before consideration for further processing.==============================

We look forward to receiving your revised manuscript.

Kind regards,

Debdutta Bhattacharya

Academic Editor

PLOS ONE

Journal Requirements:

2. Thank you for submitting the above manuscript to PLOS ONE. During our internal evaluation of the manuscript, we found significant text overlap between your submission and previous work in the introduction. We would like to make you aware that copying extracts from previous publications, especially outside the methods section, word-for-word is unacceptable. In addition, the reproduction of text from published reports has implications for the copyright that may apply to the publications.

Please revise the manuscript to rephrase the duplicated text, cite your sources, and provide details as to how the current manuscript advances on previous work. Please note that further consideration is dependent on the submission of a manuscript that addresses these concerns about the overlap in text with published work.

We will carefully review your manuscript upon resubmission and further consideration of the manuscript is dependent on the text overlap being addressed in full. Please ensure that your revision is thorough as failure to address the concerns to our satisfaction may result in your submission not being considered further.

Reviewers' comments:

Reviewer's Responses to Questions

**Comments to the Author**

1. Is the manuscript technically sound, and do the data support the conclusions?

Reviewer #1: Yes

Reviewer #2: Partly

2. Has the statistical analysis been performed appropriately and rigorously? 

Reviewer #1: No

Reviewer #2: No

3. Have the authors made all data underlying the findings in their manuscript fully available?

Reviewer #1: Yes

Reviewer #2: Yes

4. Is the manuscript presented in an intelligible fashion and written in standard English?

Reviewer #1: No

Reviewer #2: No

5. Review Comments to the Author

Reviewer #1: This study was done to assess the Sero-prevalence of anti-SARS-CoV-2 specific antibodies in vaccinated and vaccine naïve adults in Nigeria. The manuscript is clear and well organized. Understanding the immune responses induced by SARS-CoV-2 by naturally or after vaccination is important for better assessment of protection against SARS-CoV-2 as it is essential in achieving herd immunity. Data presented in this study can be employed to help policymakers make informed decisions regarding vaccination strategies. The limitation of this study is very small sample size.

Reviewer #2: Overall the manuscript has a flow similar to an academic thesis. There are a number of typos and sentences that may need to be restructured and made clearer for a reader. The author were also not clear on how data on vaccination was collected. RT-PCR testing can only confirm an active infection or a very recent infection with SARS-CoV-2 therefore previous infections cannot be confirmed without an RT-PCR test or an antigen based RDT. Participants with no vaccination history and with detected anti-SARS-CoV-2 specific antibodies may have been asymptomatically infected. Did authors use a questionnaire to collect data on any COVID-19 related symptoms for both vaccinated and unvaccinated participants? This is not clear.

Additionally, for vaccinations it is important to have data on time points for each dose as there are data of peak immune responses and when these responses are weaned off.

Below, areas in question for the comments or correction are illustrated with apostrophe

Introduction

1. 3rd paragraph, 4th line: In this regard, Nigeria aimed to vaccin'ate' 40% of its over 200 million people

2. 3rd paragraph, 9th line: 'population) are fully vaccinated'. Include what full vaccination means, two, three or four doses?

3. 4th paragraph, 4th line: It is however important to determine the antibody responses to vaccinated and unvaccinated persons in 'order' to determine possibility of previous SARS-CoV-2 exposure or protection

4. 4th paragraph, 7th line : 'S and N proteins of SARS-CoV-2', kindly include full meanings of N and S proteins

Methodology/Materials and Methods

1. Not clear how selection of participant was done

2. How did the authors get details of the vaccinations received by enrolled participants?

3. How did authors verify previous RT-PCR tests done by participants for COVID-19

4. Consider rephrasing sentence to make it clearer if authors tested participants for SARS-CoV-2 RT-PCR: Participants with past SARS-CoV-2 infection history and those without were those confirmed positive or otherwise, by real-time reverse-transcriptase polymerase-chain reaction (RT-PCR) assay on nasopharyngeal and oropharyngeal swab specimens following the World Health Organization (WHO) guideline.

Results

1. Authors did not present data on the period between each vaccination (days? Weeks?), this may explain the statement: Unexpectedly, 23.1% and 21.4% of the participants who have had first and second doses respectively, had no protective antibody to SARS-CoV-2 and The only participant who has had the booster dose was seronegative

Discussion

1. 2nd paragraph, line 5: Consider rephrasing the sentence as number of individuals tested should not be equal to the population of a country. There are guidelines on COVID-19 testing based on symptoms and contact tracing.

2. 3rd paragraph: Consider providing information of the Nigerian situation of molecular diagnostic testing. The testing is free for the community testing approach and only travelers are charges. Molecular testing capacity in Nigeria has expanded significantly across the country and its no longer limited to reference laboratories

6. PLOS authors have the option to publish the peer review history of their article (what does this mean?). If published, this will include your full peer review and any attached files.

Reviewer #1: No

Reviewer #2: No

---

## [Author Response · Author response to Decision Letter 0]

3 Sep 2022

Dear Editor,

Kindly find below our responses to the comments raised by the Academic Editors and Reviewers.

Comment

Thank you for submitting the above manuscript to PLOS ONE. During our internal evaluation of the manuscript, we found significant text overlap between your submission and previous work in the introduction. We would like to make you aware that copying extracts from previous publications, especially outside the methods section, word-for-word is unacceptable. In addition, the reproduction of text from published reports has implications for the copyright that may apply to the publications.

Please revise the manuscript to rephrase the duplicated text, cite your sources, and provide details as to how the current manuscript advances on previous work. Please note that further consideration is dependent on the submission of a manuscript that addresses these concerns about the overlap in text with published work.

We will carefully review your manuscript upon resubmission and further consideration of the manuscript is dependent on the text overlap being addressed in full. Please ensure that your revision is thorough as failure to address the concerns to our satisfaction may result in your submission not being considered further.

Response

The authors would like to thank you sincerely for this observation. The introduction section has been revised as suggested (revision carried out is as highlighted in the revised manuscript with track changes).

Comment

Has the statistical analysis been performed appropriately and rigorously?

Reviewer #1: No

Reviewer #2: No

Response

The authors would like to state that appropriate statistical tools were used for data analysis.

Comment

Is the manuscript presented in an intelligible fashion and written in standard English?

Reviewer #1: No

Reviewer #2: No

Response

Typographical and grammatical errors have been corrected.

Comments

Introduction

1. 3rd paragraph, 4th line: In this regard, Nigeria aimed to vaccin'ate' 40% of its over 200 million people

Response

Appropriate correction has been made.

2. 3rd paragraph, 9th line: 'population) are fully vaccinated'. Include what full vaccination means, two, three or four doses?

Response

Full vaccination has been defined in the sentence.

3. 4th paragraph, 4th line: It is however important to determine the antibody responses to vaccinated and unvaccinated persons in 'order' to determine possibility of previous SARS-CoV-2 exposure or protection

Response

Appropriate correction has been made.

4. 4th paragraph, 7th line : 'S and N proteins of SARS-CoV-2', kindly include full meanings of N and S proteins

Response

S and N have been written in full.

Methodology/Materials and Methods

1. Not clear how selection of participant was done

2. How did the authors get details of the vaccinations received by enrolled participants?

Response

1. Some participants were consecutively enrolled from individuals coming for voluntary COVID-19 screening at the Biorepository Laboratory, College of Medicine, University of Ibadan. The laboratory is located in Ibadan North Local Government Area of Oyo State, Nigeria. Other participants were randomly enrolled from the general populace in Ibadan North Local Government Area of Oyo State, Nigeria.

2. Details of the type of vaccine received was obtained from the vaccination card.

3. How did authors verify previous RT-PCR tests done by participants for COVID-19

Response

The results of all the participants who had RT-PCR tests were verified from the laboratory database. However, we could not verify the responses of other participants who claimed not to have undergone any screening test for SARS-CoV-2 infection. A structured questionnaire was used to obtain this information.

4. Consider rephrasing sentence to make it clearer if authors tested participants for SARS-CoV-2 RT-PCR: Participants with past SARS-CoV-2 infection history and those without were those confirmed positive or otherwise, by real-time reverse-transcriptase polymerase-chain reaction (RT-PCR) assay on nasopharyngeal and oropharyngeal swab specimens following the World Health Organization (WHO) guideline.

Response

The sentence has been revised as suggested.

Results

1. Authors did not present data on the period between each vaccination (days? Weeks?), this may explain the statement: Unexpectedly, 23.1% and 21.4% of the participants who have had first and second doses respectively, had no protective antibody to SARS-CoV-2 and The only participant who has had the booster dose was seronegative

Response

Discussion on this result has been revised.

Discussion

1. 2nd paragraph, line 5: Consider rephrasing the sentence as number of individuals tested should not be equal to the population of a country. There are guidelines on COVID-19 testing based on symptoms and contact tracing.

Response

The sentence has been revised appropriately.

2. 3rd paragraph: Consider providing information of the Nigerian situation of molecular diagnostic testing. The testing is free for the community testing approach and only travelers are charges. Molecular testing capacity in Nigeria has expanded significantly across the country and its no longer limited to reference laboratories

Response

The sentence has been revised appropriately.

---

## [Decision Letter · Decision Letter 1]

26 Dec 2022

Seroprevalence of anti-SARS-CoV-2 specific antibodies in vaccinated and vaccine naïve adult Nigerians

PONE-D-22-18914R1

Dear Dr. Rahamon,

We’re pleased to inform you that your manuscript has been judged scientifically suitable for publication and will be formally accepted for publication once it meets all outstanding technical requirements.

Kind regards,

Debdutta Bhattacharya

Academic Editor

PLOS ONE

Additional Editor Comments (optional):

Reviewers' comments:

Reviewer's Responses to Questions

**Comments to the Author**

1. If the authors have adequately addressed your comments raised in a previous round of review and you feel that this manuscript is now acceptable for publication, you may indicate that here to bypass the “Comments to the Author” section, enter your conflict of interest statement in the “Confidential to Editor” section, and submit your "Accept" recommendation.

Reviewer #2: All comments have been addressed

2. Is the manuscript technically sound, and do the data support the conclusions?

Reviewer #2: Yes

3. Has the statistical analysis been performed appropriately and rigorously? 

Reviewer #2: Yes

4. Have the authors made all data underlying the findings in their manuscript fully available?

Reviewer #2: Yes

5. Is the manuscript presented in an intelligible fashion and written in standard English?

Reviewer #2: Yes

6. Review Comments to the Author

Reviewer #2: The authors have adequately addressed the weaknesses identified earlier and the manuscript is much improved as a result.

7. PLOS authors have the option to publish the peer review history of their article (what does this mean?). If published, this will include your full peer review and any attached files.

Reviewer #2: No

---

## [Editor Report · Acceptance letter]

12 Jan 2023

PONE-D-22-18914R1 

Seroprevalence of anti-SARS-CoV-2 specific antibodies in vaccinated and vaccine naïve adult Nigerians 

Dear Dr. Rahamon:

I'm pleased to inform you that your manuscript has been deemed suitable for publication in PLOS ONE. Congratulations! Your manuscript is now with our production department. 

Kind regards, 

on behalf of

Dr. Debdutta Bhattacharya 

Academic Editor

PLOS ONE